# Study on the Characteristics of Size-Segregated Particulate Water-Soluble Inorganic Ions and Potentially Toxic Metals during Wintertime in a High Population Residential Area in Beijing, China

Kai Xiao [1], Ao Qin [1], Weiqian Wang [1], Senlin Lu [2] and Qingyue Wang [1,*]

[1] Graduate School of Science and Engineering, Saitama University, 255 Shimo-Okubo, Sakura-ku, Saitama 338-8570, Japan; xiao.k.662@ms.saitama-u.ac.jp (K.X.); benben19902002@126.com (A.Q.); weiqian@mail.saitama-u.ac.jp (W.W.)

[2] School of Environmental and Chemical Engineering, Shanghai University, 99 Shangdalu, Baoshan District, Shanghai 200444, China; senlinlv@staff.shu.edu.cn

* Correspondence: seiyo@mail.saitama-u.ac.jp; Tel.: +81-(48)-8583733

**Abstract:** Airborne particulate matter (PM) pollution often occurs in the wintertime in northern China, posing a potential threat to human health. To date, there are limited studies about the metals and inorganic ions to link source apportionments and health risk assessments in the different size-segregated PM samples. In this study, our samples were collected by a high-volume air sampler from 26 December 2018 to 11 January 2019 in a high population residential area (Beijing). Water-soluble inorganic ions, metal elements in the different size-segregated PM samples were determined for health risk assessments by inhalation of PM. During the sampling period in Beijing, the average concentrations of $PM_{1.1}$ and $PM_{1.1-2.0}$ were $39.67 \pm 10.66$ µg m$^{-3}$ and $32.25 \pm 6.78$ µg m$^{-3}$. Distinct distribution profiles characterized the different elements. The markers of coal combustion Pb, As, and Se had >52% of their mass concentration in $PM_{1.1}$. The average mass ratios of $(NO_3^- + NO_2^-)/SO_4^{2-}$, $Cl^-/Na^+$, $Cl^-/K^+$, and $Cl^-/(NO_3^- + NO_2^-)$ were 1.68, 6.58, 6.18, and 0.57, which showed that coal combustion and vehicle emissions were the main anthropogenic sources of PM in Beijing in winter. $PM_{1.1}$ was the major contributor of Pb, Cd, and As for carcinogenic risks (CR) and hazard quotient (HQ). It was indicated that $PM_{1.1}$ is more harmful than coarse PM. The toxic elements of Cr (VI) $(1.12 \times 10^{-6})$, V $(0.69 \times 10^{-6})$, and As $(0.41 \times 10^{-6})$ caused higher CR for children than Ni, Cd, Co, and Pb. Meanwhile, Pb $(35.30 \times 10^{-6})$ and Ni $(21.07 \times 10^{-6})$ caused higher CR for adults than As, Cr (VI), V, Co, and Cd, especially $PM_{1.1}$. This study provides detailed composition data and the first report on human health in a high population residential area in Beijing.

**Keywords:** airborne particulate matter; metal elements; water-soluble inorganic ion; anthropogenic sources; cancer risk

## 1. Introduction

In recent decades, with the rapid economic growth and urbanization of China, air pollutants have rapidly increase to become an environmental issue of public health concern in most metropolitan areas in China [1,2]. According to the records in the scientific literature, the significant risks posed to human health by the inhalation of airborne particles (Particulate Matter, PM) are strongly associated with their size and physico-chemical characteristics [3–5]. PM can be classified as coarse particles ($PM_{10}$ with an aerodynamic diameter less than 10 µm), fine particles ($PM_{2.5}$), and especially sub-micrometer ($PM_1$) and ultrafine particles (UFPs, $PM_{0.1}$). PM can be derived from both primary sources, in which they are directly produced by a series of human activities (e.g., fossil fuel combustion, industrial metallurgical processes, vehicle emissions and waste incineration) into the at-

mosphere, and secondary aerosol formation, released from natural sources (e.g., soil dust, volcanism, erosion, surface winds and forest fires) [6–8].

Epidemiological and experimental studies have shown that coarse particles, in general, remain in the upper respiratory system and stay there for a long time whereas fine particles penetrate deep respiratory system and deposit in the alveolar region, entering the blood circulation system more easily, and translocating to extrapulmonary organs including the liver, spleen, heart, and even brain threatening human health [9]. Potentially toxic metals, which are cytotoxic, concealed, persistent, and biologically accumulated, play a decisive role in the assessment of atmospheric pollution and the hazards to human health [10]. After breathing into the human body, they may cause various human dysfunctions or cause a variety of diseases: As, Cr, Ni, Pb, and Cd have certain carcinogenic ability, while As and Cd have potential teratogenic effects on the human body, and Pb and Hg are toxic to the fetus [11,12]. Cr, Ni, Cu, Fe, Co, Mn, As, V, and Zn could support electron exchange [13] and induce the formation of reactive oxygen species (ROS) in the lungs [14], causing damage of oxidative DNA and inflammation of respiratory tracts [15]. Lead (Pb) is a well-known toxic element that may be harmful to the nervous and hematopoietic system, leading to impaired growth and mental function [16]. Relevant literature shows that about 70–80% of the metal elements in the atmosphere are adsorbed on fine particles [17]. World Health Organization (WHO) estimated that air pollution was associated with approximately 300,000 premature deaths per year in China [18]. Consequently, the presence of different PM-bound toxic compounds may pose severe health concerns and information about their size-distribution is of primary relevance to determine and quantify the potential deleterious effects on human health.

Beijing is the capital city of China with a population of approximately 21.54 million and annual coal consumption of 17.62 million tons in 2018 [19,20]. Beijing as one of the core cities in Beijing-Tianjin-Hebei (BTH) (high elements emission areas) [21], it has been troubled by air pollution. Due to heating and meteorological conditions, air pollution in the wintertime is typically more serious than in other seasons in Beijing [22]. In the past few decades, many studies have investigated the characteristics and sources of atmospheric PM in Beijing. In $PM_{2.5}$ and $PM_{10}$ samples from Beijing, different elements showed different size distributions during summer and winter [2]. Further, 13 elements in $PM_{2.5}$ in urban Beijing were measured were to investigate the concentration of elements, illustrate their temporal variations, and estimate the health risks [23]. The concentrations of metals and ions, their characteristics, and a comparison in hazy and non-hazy days of $PM_{10}$ were discussed [1]. There are few studies on sources analysis and health risks of metal elements in size-segregated PM in Beijing, many studies have assessed human health risk caused by individual sizes (often for either $PM_{10}$ or $PM_{2.5}$) [1,2,18,24]. The compounds of PM are complex and have obvious seasonal and regional differences [25]. In addition, due to changes in physical and chemical composition, the toxic effects of particulate matter vary greatly with geographic location. Therefore, it is very important to quantify the chemical composition of aerosols to determine the potential deleterious effects on human health, especially in higher population residential area. Our sampling site is in a high population residential area (Figure 1), with about 600,000 people, and the traffic volume is very heavy in the morning and evening rush hours. Residents living in this area are potential receptors for metals in the air.

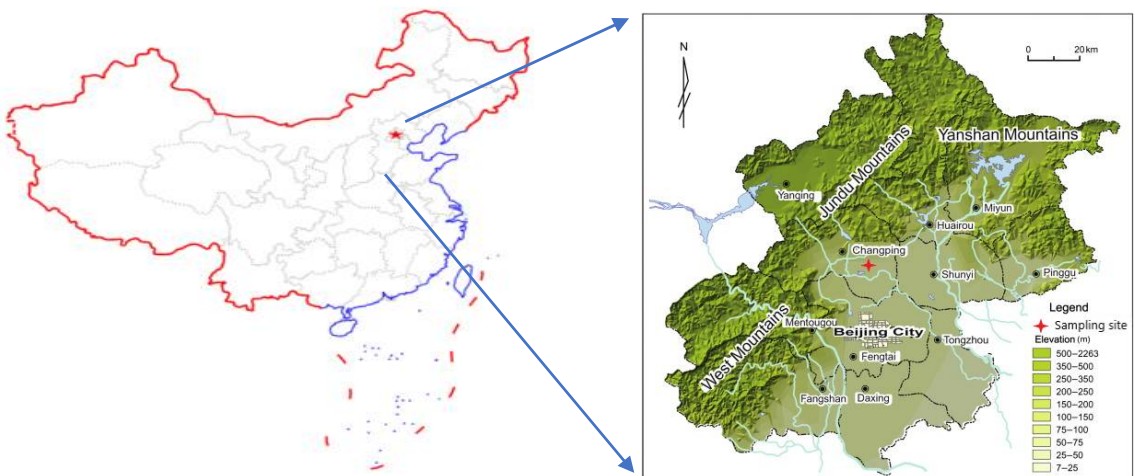

**Figure 1.** Map (**left**) of the sampling area, the map (**right**) indicates the sampling site in Beijing.

Given the background discussion above, in this study, we collected atmospheric samples with a high-volume air sampler (Anderson Sampler, HV-RW, Shibata Science Co., Ltd., Japan) in a high population residential area of Beijing during the winter (26 December 2018 to 11 January 2019). Nine water soluble inorganic ions ($Cl^-$, $NO_3^-$, $NO_2^-$, $SO_4^{2-}$, $NH_4^+$, $Na^+$, $K^+$, $Ca^{2+}$, and $Mg^{2+}$) were analyzed by Ion Chromatography (IC, ICS1600, Dionex Aquion, Thermo Fisher Scientific CO, Waltham, MA, USA), the mass concentration of 21 metal elements was measured by inductively coupled plasma mass spectrometry (ICP-MS, Agilent 7700, Agilent Technologies, Inc., Santa Clara, CA, USA). The main objectives of this study were: (1) to investigate the occurrence levels of metals elements, water soluble inorganic ions and their size distributions, (2) to identify the potential sources contributing to enrichment of metals, and (3) estimate the health risk to child and adults of several toxic trace elements (As, Cd, Cr (VI), V, Ni, Co, and Pb).

## 2. Materials and Methods

### 2.1. Sites and Sampling

As the capital of China, Beijing is the political, economic, and cultural center of China. Beijing is located at the northwestern edge (39.4–41.6 N°, 115.7–117.4 E°) of the North China Plain, adjacent to Tianjin to the east, and the other adjacent to Hebei Province, surrounded by the Taihang and Yanshan Mountains in the west, north and northeast. Moreover, it is one of the most crowded cities in the world [26]. The resident population of Beijing city is 21.54 million and car ownership of 6.36 million vehicles in 2018 [19,20]. This area experiences a monsoon-influenced humid continental climate, which is characterized by hot, humid summers, and cold, dry winters [27]. The mean temperature over the year is 11.9 °C, with annual precipitation of 400–500 mm [28].

The aerosol sampling was conducted by a high-volume air sampler at a flow rate of 566 L/min in Beijing from 26 December 2018 to 11 January 2019. The sampler (Anderson Sampler, equipped with 5 cut particle sizes: <1.1, 1.1–2.0, 2.0–3.3, 3.3–7.0, >7.0 μm can collect the particulate matter (PM) in the flue gas on five quartz filter membranes according to the aerodynamic diameters such as $PM_{1.1}$, $PM_{1.1–2.0}$, $PM_{2.0–3.3}$, $PM_{3.3–7.0}$, $PM_{>7.0}$, respectively. The quartz filter used for collecting PM was baked in a muffle furnace (450 °C) for 6 h before sampling and placed in a constant temperature and humidity chamber at 25 °C and 45% humidity for 24 h and then wrapped in clean aluminum foil paper and placed in a refrigerator at −45 °C until use. After sampling, the membranes were equilibrated and weighed again using the same procedure. To ensure accuracy, each filter was weighed at least three times before and after sampling, and the results were averaged. After weighing, store the filter at −45 °C until analysis. The sampling duration was 47 h and was changed to new quartz filter at about 12:00 a.m. During the representative sampling period, the

temperature (T; °C), relative humidity (RH; %), wind speed (WS; km/h), and wind direction (WD) and other atmospheric pollutants ($SO_2$, CO, $NO_2$ and $O_3$) were collected from the website of Air quality online monitoring and analysis platform (https://www.aqistudy.cn; accessed on 20 August 2020) and Ventusky- Weather Maps (https://www.ventusky.com; accessed on 20 August 2020).

### 2.2. Potentially Toxic Metals Characterization via ICP-MS

Inductively coupled plasma mass spectrometry (ICP-MS) was used to identify 21 elemental species (Na, Mg, Al, K, Ca, Ti, V, Cr, Mn, Fe, Co, Ni, Cu, Zn, As, Se, Sr, Cd, Sb, Ba, Pb) concentration [29]. After weighing, a certain amount of filter was cut and placed in polytetrafluoroethylene (PTFE) reaction tubes, then each filter was digested with a 5:2:1 mixture of $HNO_3$-$H_2O_2$-HF in PTFE vessels and heated in a microwave system (ETHOS UP, MAXI-44). There were six heating steps for microwave digestion of filter. (1) The temperature was increased from room temperature to 90 °C and was maintained for 3 min. (2) The temperature was increased from 90 to 150 °C and was maintained for 5 min. (3) The temperature was increased from 150 to 175 °C and was maintained for 17 min. (4) The temperature was increased from 175 to 200 °C and was maintained for 30 min. (5) After cooling for 30 min, the polypropylene reaction vessel was removed from the microwave sample pretreatment device, and the decomposition solution in the PTFE reaction vessel put into the polypropylene digestion tube. (6) The polypropylene digestion tube was placed in the acid decomposition system and heated at 100 °C for about 5 h to remove hydrofluoric acid until the solution is about 0.1 mL. The digested solution was then diluted to 10 mL with 2% $HNO_3$ and stored at room temperature until analysis. Three blank filters were treated and analyzed with the same methods as for and metals to obtain background values. Potentially toxic metals were analyzed by ICP-MS at the Center for Environmental Science in Saitama (CESS) in Japan.

### 2.3. Water-Soluble Inorganic Ions

An ultrasonic method was used to extract water-soluble inorganic ions from portions of the PM filter samples, and normally over 98% of sulfate, nitrate, and ammonium can be extracted. The filter was submerged in a vial with 10 mL ultrapure water, sealed and subjected to ultrasound for 20 min for each extraction. The extraction was repeated 3 times. The extract was filtered with a 0.22 μm PTFE and then the concentration of water-soluble inorganic ions ($Cl^-$, $NO_3^-$, $NO_2^-$, $SO_4^{2-}$, $NH_4^+$, $Na^+$, $K^+$, $Ca^{2+}$, and $Mg^{2+}$) were analyzed by ion chromatography to determine the concentrations of water-soluble inorganic ions [1,29].

### 2.4. The Crustal Enrichment Factors (CEFs)

To assess the contribution of anthropogenic sources and crustal origin of various trace elements in multi-size PM. In this study, the crustal enrichment factors (CEFs) of all elements were calculated by dividing the relative abundance in the PM sample by the average abundance in the upper continental crust. (CEFs) were calculated by Equation (1) [30].

$$\text{CEFs} = (\text{Eatm}/\text{Ratm})/(\text{Ecrust}/\text{Rcrust}) \qquad (1)$$

where E and R represent the investigated element and reference element for crust material, respectively. (Eatm/Ratm) is the concentration ratio of E to R in PM sample while (Ecrust/Rcrust) is the concentration ratio of E to R from earth upper continental crust [31]. To facilitate interpretation, classification of EF follows: similar to crust (CEFs < 1), the elements almost all originated from the crust; low enrichment (CEFs 1–10), the element mainly contributed by natural sources while slightly contributed by anthropogenic emissions; moderate enrichment (CEFs 10–100), the elements released from human activities and highly enrichment (CEFs > 100), the elements affected by human activities in most of the studies [4,32,33]. In the present study Al, high abundance in the Earth's crust composition, was selected as the reference element [29].

### 2.5. Health Risk Assessment

In this study, based on the US-EPA Integrated Risk Information System (IRIS) and International Agency for Research on Cancer (IARC), the metals can be divided into carcinogens and non-carcinogens. Group 1 (carcinogenic to humans): As, Cd, Cr (VI) and Ni; Group 2A (probably carcinogenic to humans); Group 2B (possibly carcinogenic to humans): Co, V, and Pb. The concentration of Cr (VI) was presumed to be 1/7 of the total concentration of Cr when calculating the Cr health risk. According to US-EPA Region RSL (Regional Screening Levels), the Cr (VI) to Cr (III) ratio is 1:6 [9,34]. We analyzed the health risks of these seven toxic metal elements (As, Cd, Cr (VI), Ni, Co, V, and Pb).

To determine the probability of non-carcinogenic and carcinogenic risks (CR) to the public due to PM-bound metals, the risk to human health should be assessed. Fine particle matter through inhalation would deposit into the alveolar region. However, the alveolar region does not have protective mucus layers, so particles deposited into this region are difficult to eliminate, causing considerable health risks for humans [35]. Moreover, the alveolar area is difficult to eliminate particles deposited in this area, it can be posing a considerable health risk to the human body [36]. Thus, particles deposited in the alveolar region via inhalation route are considered to play the most important role in threatening human health [37].

$$EC = \frac{(C_i \times ET \times EF \times ED)}{ATn} \tag{2}$$

$$HQ = \frac{EC}{(RfC \times 1000)} \tag{3}$$

$$CR = UR \times EC \tag{4}$$

The carcinogenic risks (CR) are the likelihood that an individual will develop any type of cancer after being exposed to cancer risk during their lifetime. The sum of the hazard quotient (HQ, index of non-carcinogenic risk) was applied to evaluate the overall potential for non-carcinogenic effects caused by more than one chemical. Thus, both the noncarcinogenic and carcinogenic health risks of metals in PM via inhalation pathways for each element can be calculated as Equations (2)–(4) [38]:

The exposure concentration (EC) of each element through inhalation ($\mu$g m$^{-3}$). where $C_i$ is the average concentration of individual metals of PM size i. where ET, EF, ED and ATn are exposure time (h day$^{-1}$), exposure frequency (days year$^{-1}$), exposure duration (year), and average lifetime (hours). The explanations and values of all the parameters are listed in Table 1.

**Table 1.** Parameters and values used in the human health risk.

| Parameter | Definition | Unit | Value | | References |
|---|---|---|---|---|---|
| | | | **Child** | **Adult** | |
| Exposure time | ET | h day$^{-1}$ | 24 | 24 | [39] |
| Exposure frequency | EF | days year$^{-1}$ | 350 | 350 | [12] |
| Exposure duration | ED | year | 6 | 24 | [36] |
| Average lifetime | ATn | hours | non-carcinogenic AT = ED × 365 (day year$^{-1}$) × 24 (hour day$^{-1}$) carcinogenic AT = 80 × 365 (day year$^{-1}$) × 24 (hour day$^{-1}$) [40] | | |

UR is the inhalation unit risk of trace metal ($\mu$g·m$^{-3}$)$^{-1}$, RfC is the corresponding reference concentration for the non-carcinogenic trace metal (mg·m$^{-3}$). The HQ value higher than 1 implies that the non-cancer risk merits attention. Carcinogenic risk is the probability that an individual will develop cancer because of exposure to carcinogenic hazards over the individual's lifetime. CR value is lower than $1 \times 10^{-6}$ (i.e., one additional case of cancer per million people), indicating acceptable risk. The acceptable or tolerable

risk for regulatory purposes is defined as a carcinogenic risk of between $1 \times 10^{-6}$ and $1 \times 10^{-4}$ [41]. The parameter of UR and RfC value of potentially toxic metals are shown in Table 2.

**Table 2.** The UR and RfC value of some potentially toxic metals.

| Element | Unit Risk (UR) $(\mu g\ m^{-3})^{-1}$ | References | RfC $(mg\ m^{-3})$ | References |
|---|---|---|---|---|
| V | $8.3 \times 10^{-3}$ | PPRTVs [a] | $1.0 \times 10^{-4}$ | |
| Cr$^{6+}$ | $1.2 \times 10^{-2}$ | RAIS [b] | $1.0 \times 10^{-4}$ | USEPA [d] |
| Co | $9.0 \times 10^{-3}$ | PPRTVs [a] | $6.0 \times 10^{-6}$ | RAIS [b] |
| Ni | $2.4 \times 10^{-4}$ | RAIS [b] | $1.4 \times 10^{-5}$ | CALEPA [c] |
| As | $4.3 \times 10^{-3}$ | RAIS [b] | $1.5 \times 10^{-5}$ | CALEPA [c] |
| Cd | $1.8 \times 10^{-3}$ | RAIS [b] | $1.0 \times 10^{-5}$ | USEPA [d] |
| Pb | $1.2 \times 10^{-5}$ | CALEPA [c] | | |

[a] PPRTVs: Provisional Peer-Reviewed Toxicity Values (n.d.), from https://www.epa.gov/pprtv/provisional-peer-reviewed-toxicity-values-pprtvs-assessments; accessed on 15 August 2020. [b] RAIS: The Risk Assessment Information System. (n.d.), from https://rais.ornl.gov/cgi-bin/tools/TOX_search?select=chemtox; accessed on 15 August 2020. [c] CALEPA: California Environmental Protection Agency. (n.d.), from https://calepa.ca.gov/; accessed on 15 August 2020. [d] USEPA: United States Environmental Protection Agency. (n.d.), from https://www.epa.gov/iris; accessed on 15 August 2020.

*2.6. Statistics*

Statistical analysis was accomplished by Excel (2010) (Microsoft Corporation., Redmond, WA, USA) and Origin 8.0 (OriginLab Corporation., Northampton, MA, USA) was used to draw the figures in this article.

**3. Results and Discussion**

Generally, PM can be classified as coarse particles (PM$_{10}$ with an aerodynamic diameter less than 10 μm), fine particles (PM$_{2.5}$) [42,43]. In our research, in order to facilitate statistical analysis, we regard particles <1.1 μm and 1.1–2.0 μm as fine particles (with an aerodynamic diameter less than 2.0 μm), and others as coarse particles PM$_{>2.0}$ including (2.0–3.0 μm, 3.3–7.0 μm, >7.0 μm). The sum of all particles is called total suspended particles (TSP).

*3.1. Weather Conditions*

Many studies have reported that the mass concentration of size distribution varies in different conditions [43,44].The characteristics of atmospheric aerosols can be investigated based on the weather conditions [43] (T, RH, WS, and WD and atmospheric pollutants (SO$_2$, CO, NO$_2$ and O$_3$) [29].

Detailed information about the meteorological parameters, weather conditions and atmospheric pollutants was provided in the supplementary material (Tables S1–S3). During the sampling period, the temperature varied from −8 °C to 3 °C with a mean value of −2.18 ± 2.55 °C; the ambient RH was in a range of 10–30% and averaged at 18.24 ± 7.06%; the average wind speed of 4.12 ± 2.40 km h$^{-1}$. Table 1 and Figure 2 show that the lowest PM$_{2.0}$ and PM$_{>2.0}$ (38.52 and 111.41 μg m$^{-3}$, respectively), which was collected between 28–30 December, during this sampling period, the ambient temperature (−5 °C) and relative humidity (10%) was lower than other groups, and wind speed was higher than other sampling time. Meanwhile, PM$_{2.0}$ and PM$_{>2.0}$ was highest (75.95 and 207.23 μg m$^{-3}$, respectively), which was collected between 1–3 January, during the sampling period, the ambient temperature (−1.33 °C) and relative humidity (26.67%) are relatively higher, while the wind speed (3.00 km/h) is lower than most other sampling groups. Pearson correlation coefficients show that the mass concentration PM$_{3.3–7.0}$ had an apparent correlation ($p < 0.01$) with T(r = 0.85) and the mass concentration PM$_{2.0–3.3}$ also had an apparent correlation ($p < 0.01$) with T(r = 0.85). From the above discussion, we conclude that the mass concentration of particulate matter is positively correlated with temperature and humidity, and negatively correlated with wind speed. Pearson correlation coefficients between mass concentration and meteorological parameters and atmospheric pollutants were presented

in Table 3. To interpret the r-value for the correlation, the following classification was adopted, as presented in Supplementary Table S4 [45].

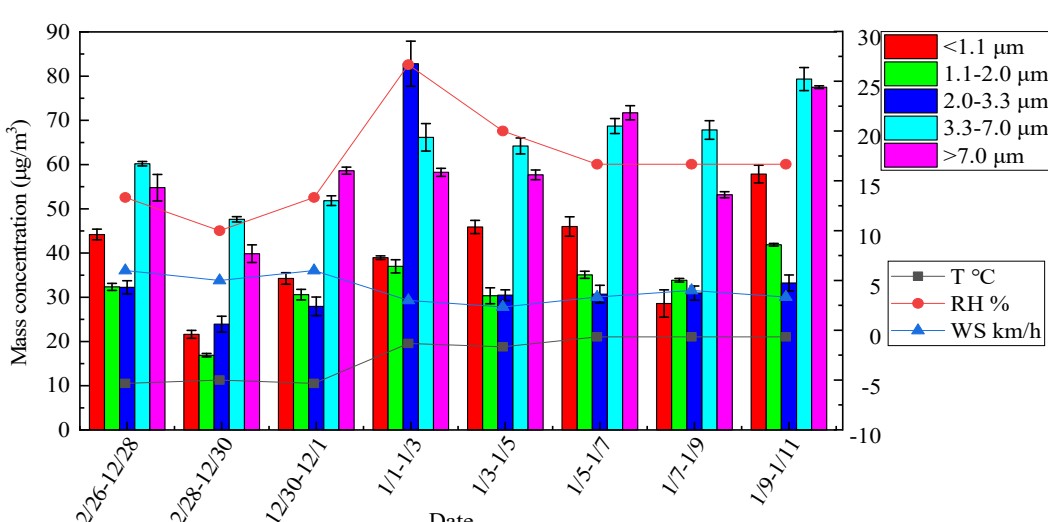

**Figure 2.** The meteorological factors and mass concentrations in different size-resolved PM during each sampling period.

**Table 3.** Pearson correlation coefficients between mass concentration and meteorological parameters and atmospheric pollutants.

| Size Range | T (°C) | RH (%) | WS (km h$^{-1}$) | SO$_2$ (µg m$^{-3}$) | CO (mg m$^{-3}$) | NO$_2$ (µg m$^{-3}$) | O$_3$-8h (µg m$^{-3}$) |
|---|---|---|---|---|---|---|---|
| <1.1 µm | 0.43 | 0.34 | −0.37 | 0.58 | 0.63 | 0.58 | −0.60 |
| 1.1–2.0 µm | 0.64 | 0.58 | −0.40 | 0.72 | 0.74 | 0.71 | −0.67 |
| 2.0–3.3 µm | 0.31 | 0.85 | −0.37 | 0.58 | 0.49 | 0.36 | −0.44 |
| 3.3–7.0 µm | 0.85 | 0.52 | −0.65 | 0.75 | 0.79 | 0.76 | −0.61 |
| >7.0 µm | 0.59 | 0.32 | −0.41 | 0.66 | 0.74 | 0.76 | −0.74 |
| <2.0 µm | 0.55 | 0.46 | −0.43 | 0.68 | 0.71 | 0.66 | −0.67 |
| >2.0 µm | 0.70 | 0.83 | −0.61 | 0.86 | 0.85 | 0.76 | −0.76 |

Pearson correlation coefficients show that the mass concentration PM$_{2.0}$ and PM$_{>2.0}$ both had an apparent correlation ($p < 0.01$) with T(r = 0.55), RH(r = 0.46), WS(r = −0.43) and T(r = 0.70), RH(r = 0.80), WS(r = −0.61), respectively; in addition, the mass concentration PM$_{2.0}$ and PM$_{>2.0}$ also presented an apparent correlation with atmospheric pollutants (PM$_{2.0}$, SO$_2$, r = 0.68, CO, r = 0.71, NO$_2$, r = 0.66, O$_3$, r = −0.67, $p < 0.01$; PM$_{>2.0}$ SO$_2$, r = 0.86, CO, r = 0.85, NO$_2$, r = 0.76, $p < 0.01$, O$_3$, r = −0.76, $p < 0.01$). This result was consistent with the study [29].

### 3.2. The Mass Concentration of Atmospheric Particulate Matter Size Distribution

In addition to weather conditions, the mass concentration distribution of aerosol particles strictly depends on their size. The size distribution of particulate matter can provide information about the chemical and physical processes. Some epidemiological evidence suggests many adverse health effects, such as chronic respiratory [46], heart disease [47], lung cancer [48], and acute respiratory infections, are closely related to exposure to PM [49,50]. Figure 2 illustrates the time series of mass concentrations in size segregated PM$_{1.1}$ in winter in Beijing. The distribution of PM concentration was bimodal, with fine particles peak PM$_{1.1}$ and coarse particles peak at PM$_{3.3–7.0}$. The detailed information about the mass concentration ratios was provided in Supplementary Table S5. The average concentrations of PM$_{1.1}$, PM$_{1.1–2.0}$, PM$_{2.0–3.3}$, PM$_{3.3–7.0}$, PM$_{>7.0}$ and TSP were 39.67 ± 10.66 µg m$^{-3}$, 32.25 ± 6.78 µg m$^{-3}$, 36.54 ± 17.70 µg m$^{-3}$, 63.24 ± 9.37 µg m$^{-3}$, and 58.95 ± 10.75 µg m$^{-3}$, and 230.06 ± 42.29 µg m$^{-3}$ respectively. The average concentration of PM$_{2.0}$, PM$_{7.0}$ and TSP in winter in Beijing, slightly exceeds the second-grade ceiling of the National Ambient

Air Quality Standard (NAAQS GB3095-1996). However, the average concentration of $PM_{2.0}$ and $PM_{7.0}$ were nearly 1.03 times and 2.45 times higher than GB3095-2012 of China for $PM_{2.5}$ (35 μg m$^{-3}$) and $PM_{10}$ (μg m$^{-3}$), respectively. The average concentration of $PM_{2.0}$ and $PM_{7.0}$ also exceed The World Health Organization (WHO) air quality guidelines more than 3.19 times and 7.58 times. It was indicated that the air pollution is much serious in winter in Beijing.

There is no large pollution source on the sampling site. However, both $PM_{2.0}$ and $PM_{7.0}$ average concentrations in winter are similar to previous research in Beijing in winter($PM_{2.5}$ 85.47 μg m$^{-3}$, $PM_{10}$ 107.20 μg m$^{-3}$) and higher in summer ($PM_{2.5}$ 60.20 μg m$^{-3}$, $PM_{10}$ 46.3 μg m$^{-3}$) [2], which are attributable to the coal combustion from residents for living and heating, vehicle emissions, road dust or uncovered ground, construction site, as well as air pollution transported from Tianjin-Hebei-Shandong province. These activities in the province are the frequent occurrence of unfavorable atmospheric diffusion conditions, resulting in increased air pollutant emissions [2].

Furthermore, the average ratios of $PM_{1.1}$/TSP, $PM_{1.1-2.0}$/TSP, $PM_{2.0-3.3}$/TSP, $PM_{3.3-7.0}$/TSP and $PM_{>7.0}$/TSP were 0.17 ± 0.03, 0.14 ± 0.01, 0.16 ± 0.05, 0.27 ± 0.03 and 0.26 ± 0.02 respectively, which indicates that fine particles significantly contribute to atmospheric particulate matter pollution in winter in Beijing. This was also observed in previous studies [8,18]. It is generally acknowledged that primary source like road dust and soil as the main emission source of coarse particulate matter, while fine atmospheric particulate matters are emitted from both primary source and secondary formation due to complex chemical processes in the atmosphere [2].

### 3.3. The Mass Concentration of Water-Soluble Inorganic Ionic Species Size Distribution and Sources

Detected ion water-soluble inorganic ionic species (WSIIs), an important of atmospheric particulate matters, which can be used to infer the chemical properties and origin of aerosols [29,39]. Figure 3 illustrated the size distribution of $Cl^-$, $NO_3^-$, $NO_2^-$, $SO_4^{2-}$, $NH_4^+$, $Na^+$, $K^+$, $Ca^{2+}$ and $Mg^{2+}$ detected in winter. The percentage (%) of water-soluble inorganic ionic species for size fraction mass concentrations in Beijing were provided in Figure 4. The ratio of $NO_3^-$/$SO_4^{2-}$, $Cl^-$/$Na^+$, $Cl^-$/$K^+$, and $Cl^-$/$NO_3^-$ in different size-that secondary pollution in the atmosphere in winter was strong. $SO_4^{2-}$ showed a slightly bimodal size distribution, with fine particles, $PM_{1.1}$, thus demonstrating a unimodal distribution of fine particles. Most of the $SO_4^{2-}$, $NO_3^-$, and $NH_4^+$ mass concentration were concentrated in fine particles, $PM_{1.1}$, which is consistent with previous studies [1,40,41]. The average mass concentration of $NO_3^-$ in the fine particles accounted for 74.94% of the total nitrate mass, while that for $NH_4^+$ was 71.85% and that for $SO_4^{2-}$ was 61.70% in winter in Beijing. $Na^+$ and $Mg^{2+}$ were bimodal, the major peak in the size range of $PM_{1.1}$ while the minor peak in the size range of $PM_{>7.0}$, meanwhile $Ca^{2+}$ was unimodal, peak of $PM_{>7.0}$. The size distribution of $K^+$, $Cl^-$, and $NO_2^-$ were bimodal, with a fine particles peak at $PM_{1.1}$ and coarse particles peak of $PM_{>7.0}$.

It was well known that that $SO_2$ and $NO_2$, which are the gaseous precursors of $SO_4^{2-}$ and $NO_3^-$, so the ratio of $NO_3^-$/$SO_4^{2-}$ could be used to compare the contribution of stationary (such as coal burning) and mobile source (such as motor vehicle exhausts) of $SO_2$ and $NO_2$ [2]. $NO_2^-$ concentration is very low, and it is unstable due to being easily oxidized by ozone, hydroxyl radicals, and hydrogen peroxide, so put $NO_2^-$ and $NO_3^-$ together to discuss. According to the previous studies show that the higher ($NO_2^-$ + $NO_3^-$)/$SO_4^{2-}$ values to mobile source over the stationary source of atmospheric pollutants. The average mass ratio of ($NO_2^-$ + $NO_3^-$)/$SO_4^{2-}$ was 1.68. Compared to the previous results, which reported the measured ratio of 0.71 during 2001–2003 [49], 1.03 in 2012 [1], between 1.31–1.16 during 2014–2015 [1], and 3.12 in 2017 [25], indicating that air pollution during the research period was mainly from the mobile source in Beijing in winter.

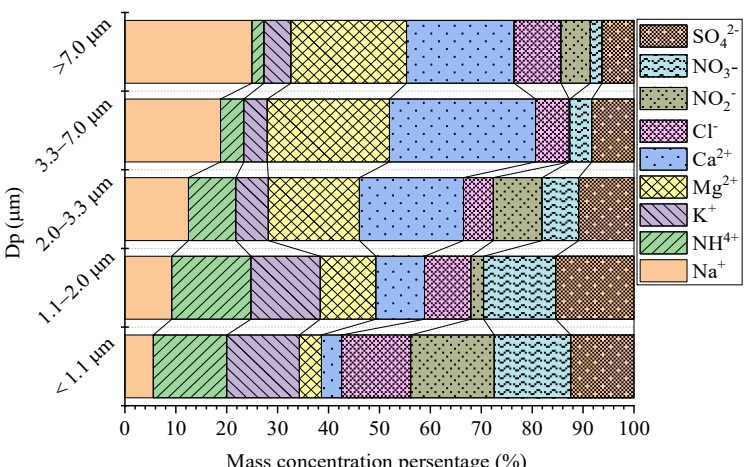

**Figure 3.** Size distribution of water-soluble inorganic ionic species.

**Figure 4.** The percentage (%) of water-soluble inorganic ionic species for size fraction mass concentrations in Beijing.

Most of the $Na^+$, $Ca^{2+}$, and $Mg^{2+}$ in coarse particles were found to be high, which are mainly from crustal source, such as re-suspended road dust, soil dust and building dust [29]. $K^+$ in fine particles is known as an inorganic tracer of biomass burning emissions [41]. The results show that the distribution of mass concentration of $Na^+$, $Ca^{2+}$, and $Mg^{2+}$ were consistent with the distribution of mass concentration of Na, Mg, and Ca (Figure 5). The dominant source of $Cl^-$ was generally considered from coal burning, biomass burning and vehicle exhaustion fine fraction and sea water in a coarse fraction [1]. The linear relationship between $K^+$ and $Cl^-$ was strong (R > 0.98). Meanwhile, the linear relationship between $Na^+$ and $Cl^-$ was weak (R < 0.20). The average mass ratio of $Cl^-/Na^+$, $Cl^-/K^+$ and $Cl^-/(NO_2^- + NO_3^-)$ were 6.58, 6.18 and 0.57, respectively. The above discussion indicated that the major contributor of $Cl^-$ was coal burning and the minor contributor of $Cl^-$ was biomass burning and vehicle exhaustion in Beijing during the winter.

### 3.4. Potentially Toxic Metals

Generally, PM is produced due to weathering and soil suspension, construction, coal, oil burning and resuspension of industrial dust [51,52]. Potentially toxic metals are considered to be an important ingredient in inducing respiratory diseases and cancer due to their toxicity [42,53,54]. Thus, determining the types and sources of heavy metals is a direct and effective way to control potentially toxic metals pollution.

### 3.4.1. The Mass Concentration of Potentially Toxic Metals Size Distribution

Mass concentrations of the total 21 elements (Na, Mg, Al, K, Ca, Ti, V, Cr, Mn, Fe, Co, Ni, Cu, Zn, As, Se, Sr, Cd, Sb, Ba, Pb) in multi-size PM samples were determined by using ICP-MS. Although the mass concentration of some potentially toxic metals is very low, some of these potentially toxic metals adsorbed on PM are harmful to public health and therefore cannot be ignored. The size distribution of potentially toxic metals for Beijing samples during the winter were illustrated in Figures 5–7 and Supplementary Table S6.

According to their mean mass size distribution, 21 elements could be divided into three groups: bimodal, unimodal, irregular. Group I were classified into two situations (Figure 5), the first case I a (Figure 5a): Na, Mg, Ca, Al, Ti, Fe, Ni, Cu, Mn, Sr, and V were bimodal, the highest peak in the size range sizes of $PM_{>7.0}$, however, the lowest peak in the size range of $PM_{1.1}$. (The corresponding mass concentration ratio of aggregate particle size were 50.45%, 48.56%, 47.12%, 46.59%, 49.38%, 45.20%, 34.61%, 31.33%, 37.44%, 48.76%, 51.70% vs. 26.57%, 17.42%, 16.41%, 22.53%, 17.80%, 15.77%, 23.31%, 20.75%, 24.53%, 18.18%, 16.25%). Another case I b (Figure 5b): Sb Zn and K were also bimodal, hoverer, the maximum average mass concentration aggregate particle size and the minimum aggregate particle size are opposite to the first case (the corresponding mass concentration ratios of aggregate particle size were 38.07%, 39.02%, 38.10% vs. 19.24%, 33.99%, 21.84%). Figure 6 shows that the potentially toxic metals with particle size- unimodal distribution. Metals in group II a: the mean mass concentration of Ba and Co showed a unimodal size distribution, with coarse particles peak at $PM_{>7.0}$ while Cd, As, Se and Pb (group II b) were accumulated on sizes of $PM_{1.1}$. The elements of Ba, Co appeared to have most of their mass portion in the coarser size-range of $PM_{>7.0}$, with more than 44% of the TSP (Table S6). The mean mass concentration of Cd, As, Se and Pb in sizes of $PM_{1.1}$ contributed about 67.98%, 52.15%, 78.73%, and 70.95% to the total corresponding metal in TSP, respectively. Figure 7 shows that the metal elements with particle size- irregular distribution Group III: the major of Cr was localized on sizes of $PM_{>7.0}$, the minor was accumulated on $PM_{2.0–3.3}$.

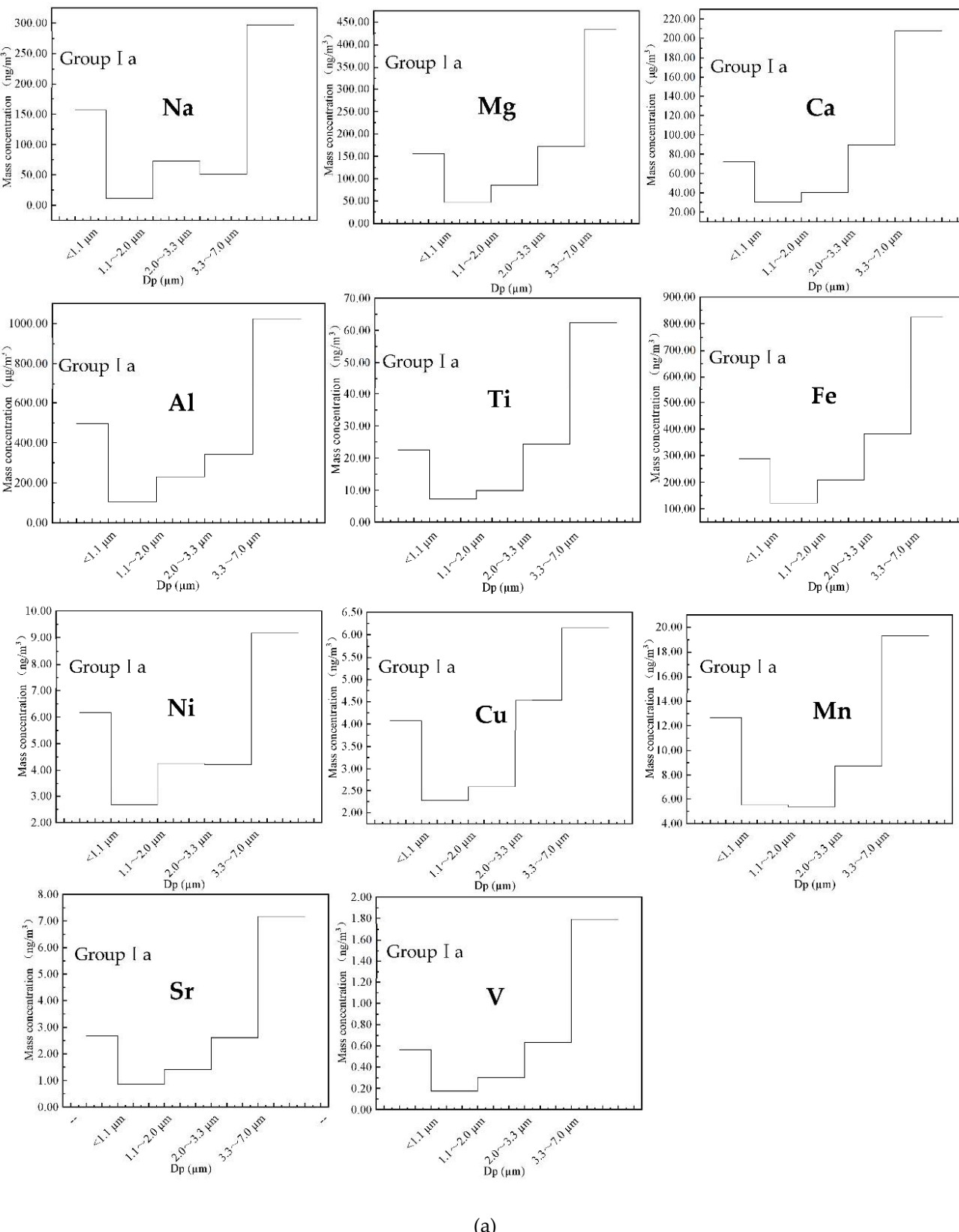

(a)

**Figure 5.** *Cont.*

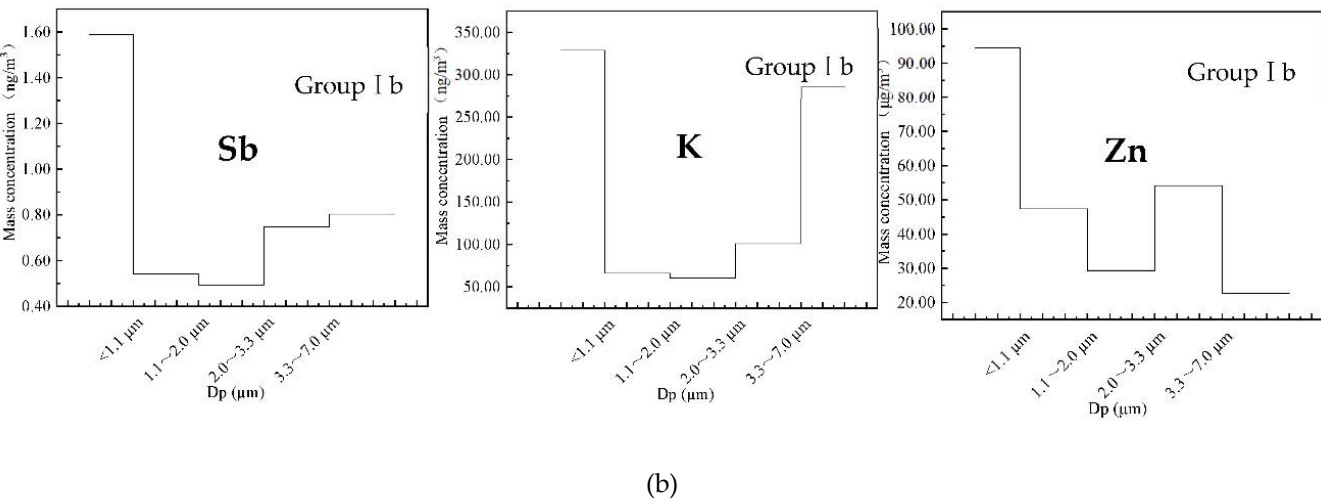

(b)

**Figure 5.** Potentially toxic metals with particle size-bimodal distribution. (**a**) Group I a is the mass concentration mainly concentrated in PM$_{>7.0}$; (**b**) Group I b is the mass concentration mainly concentrated in PM$_{1.1}$.

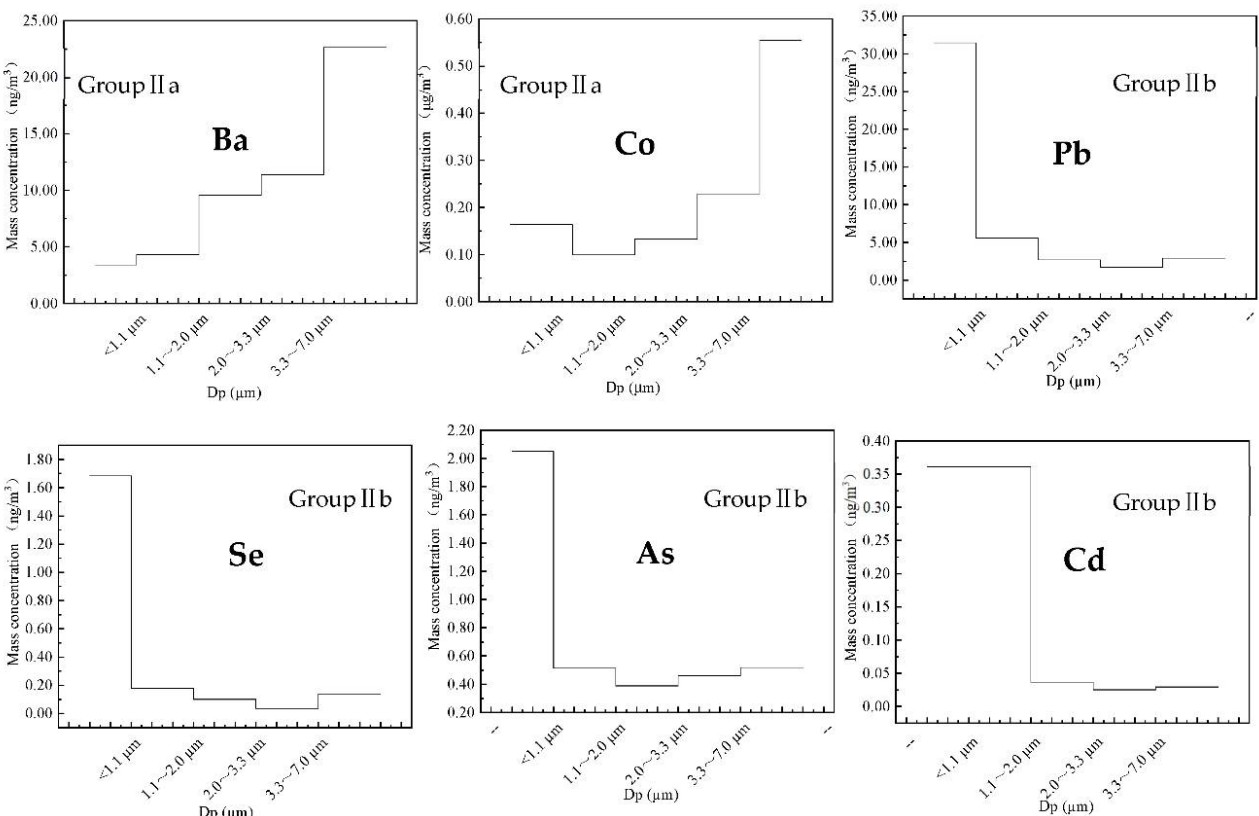

**Figure 6.** Potentially toxic metals with particle size-unimodal distribution. Group II a is the mass concentration mainly concentrated in PM$_{>7.0}$; Group II b is the mass concentration mainly concentrated in PM$_{1.1}$.

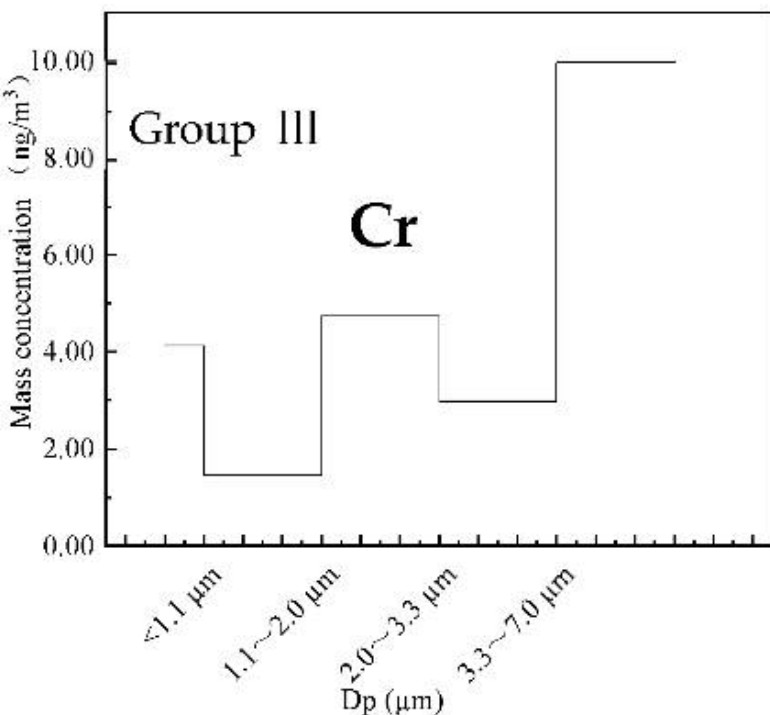

**Figure 7.** Potentially toxic metal of Cr with particle size-irregular distribution.

Total concentrations of Na, Mg, Al, Ca, and Fe in winter contributed about 80.24% and 69.15% to the total TSP and $PM_{2.0}$ element concentrations respectively, indicating that Na, Mg, Al, Ca and Fe would be more likely originated from dusts storm and road dust related sources and occurred mostly in coarse particles. The total mass concentrations of As, Cd, Pb, and Se accounted for 0.69% and 1.95% of the whole element's concentrations for TSP and $PM_{2.0}$, respectively, which indicated that As, Cd, Pb, and Se could be mainly associated with fuel combustion sources and existed largely in fine particles. This is in good accordance with the previous study [55], indicating that coal consumption of residential and other sectors for supplying heat in winter would increase emissions of Cd, Cr, and Pb. The north, west, and southwest of our study area were surrounded by mountains and largely covered by forest and scenery protection zone (see Figure 1). while the southern plain east regions were not only densely populated, but also were vulnerable affected by the pollutant's diffusion and trans-boundary transportation from surrounding regions, such as Tianjin, Hebei, and Shandong.

3.4.2. Source-Apportionment of Potentially Toxic Metals by Crustal Enrichment Factors

The crustal enrichment factors (CEFs) of each element of size-segregated particles in Beijing were illustrated in Figure 8. In this study, the CEFs values of Na, Ti, V, and Ca in five stages of all PM size and K in the range of particle size <3.3 µm were close to 1, fine PM, which suggested that these elements would be more likely originated from natural sources (including re-suspended road dust, soil dust and building dust) [56,57] and had no obvious enrichment in aerosols.

Most elements in $PM_{1.1}$ and $PM_{1.1-2.0}$ revealed higher CEFs than those in $PM_{>7}$, when the CEFs values between 1 and 10, reflecting that the metals in the smaller particles were significantly contributed by anthropogenic emissions, well consistent with previous research results [58]. The CEFs values of Mg, Ba, Co, Mn, Sr, and Fe in all particle size ranges were between 1 and 10, low enrichment, indicate a mixed contribution of metals majorly from natural sources and minorly from anthropogenic sources. Fe and Mn were speculated to be released from lubricating oil [59,60]. Fe, Mn, Ba can be released from the wear of rails or braking systems, or diesel engines from vehicles [61], while it is an

important part of the crustal elements. Here, the mean CEFs of Cu, Ni, Pb, Cr, and As were normally below 100, indicating these elements originated mainly from anthropogenic sources and had moderate enrichment in the PM. Cu was the metal with the moderate enrichment (17.48–80.05), major emitted from exhaust emissions and tire wear. The crustal enrichment factors for As and Pb were 8.39–123.39 and 13.37–551.65, respectively, they mainly came from coal combustion in metallurgy, thermal power and other industries [48]. Leaded gasoline was eliminated more than ten years ago, but as this research shows, lead is still ubiquitous in the environment. Cr is widely used in the electroplating and leather tanning industries [24], and a large number of leather industries and metal electroplating industries were found near Beijing [39], which verified that the interpretation of this source was reasonable. The CEFs values of Sb, Se, Zn, and Cd were greater than 100 in all PM sizes, which suggested that these elements were highly enrichment and significantly affected by vehicle emissions and industrial sources or coal combustion, including metal smelting and incineration emissions [60,62]. From Figure 6, we conclude that when CEFs > 10, the CEFs values increased with increasing aerosol particle size, reflecting that metals in coarse particles were mainly contributed by human activities.

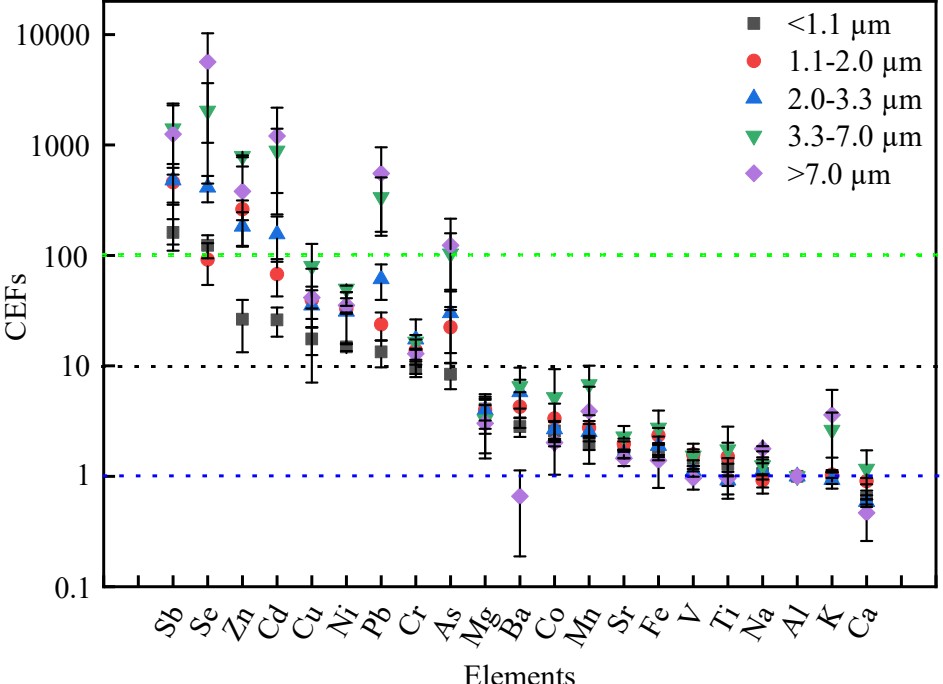

**Figure 8.** The crustal enrichment factors (CEFs) of each element of size-segregated particles in Beijing. The blue dashed line indicates that CEFs are less than 1. The black dashed line indicates that CEFs are less than 10. The green dashed line indicates that CEFs are less than 100.

### 3.4.3. Health Risk Assessment

Our sampling site is the largest residential area in Asia, with about 600,000 people, and the traffic volume is very heavy in the morning and evening rush hours. Residents living in this area are potential receptors for metals in the air. Fine and coarse atmospheric particulate matter has an important impact on human health and inhalation is the typical main route of direct exposure of toxic elements bound to PM in the atmosphere [30].

Figure 9 show the carcinogenic and non- carcinogenic risks of each toxic elements for children and adults within $PM_{1.1}$, $PM_{1.1-2.0}$, $PM_{2.0-3.3}$, $PM_{3.3-7.0}$, and $PM_{>7.0}$ in Beijing during the sampling period. As expected, most of the toxic metals exhibited high CR values in the smaller particles (<1.1 μm) because of their high deposition efficiencies [38]. The total CR values (reached $2.42 \times 10^{-6}$ for children and $66.71 \times 10^{-6}$ for adults, respectively) were exceeded the acceptable level ($1 \times 10^{-6}$), indicating that we should pay more attention to

these toxic elements [43]. Compared with previous studies, carcinogenic risks for children and adults is lower than those carcinogenic risks in Nanjing ($2.57 \times 10^{-5}$ for children and $4.59 \times 10^{-5}$ for adults) [39], in Linfen ($2.91 \times 10^{-5}$ for children and $7.75 \times 10^{-5}$ for adults) [9], In Changzhi ($2.58 \times 10^{-6}$ for children and $10.31 \times 10^{-6}$ for adults) [33], in Kanpur ($1.60 \times 10^{-5}$ for children and $3.99 \times 10^{-6}$ for adults) [63], and in Ningbo ($6.24 \times 10^{-6}$ for children and $2.50 \times 10^{-5}$ for adults) [64]. Supplementary Figure S1 show that the relative portions of carcinogenic and non- carcinogenic risks of toxic elements within $PM_{1.1}$, $PM_{1.1-2.0}$, $PM_{2.0-3.3}$, $PM_{3.3-7.0}$ and $PM_{>7.0}$ in Beijing during the sampling period. $PM_{1.1}$ was the major contributor of Pb, Cd and As for CR and HQ (Supplementary Table S7), it was indicated that $PM_{1.1}$ is more harmful than coarse PM. The toxic elements of Cr (VI) ($1.12 \times 10^{-6}$), V ($0.69 \times 10^{-6}$) and As ($0.41 \times 10^{-6}$) were caused higher CR for children than Ni, Cd, Co, and Pb. Meanwhile, Pb ($35.30 \times 10^{-6}$) and Ni ($21.07 \times 10^{-6}$) caused higher CR for adults than As, Cr (VI), V, Co, and Cd, especially $PM_{1.1}$ (Supplementary Table S8). It was indicated that V, Cr (VI), and As may be more dangerous for children, whereas Pb and Ni may be more dangerous for adults. The toxic element of Ni had the highest HQ for children and adults.

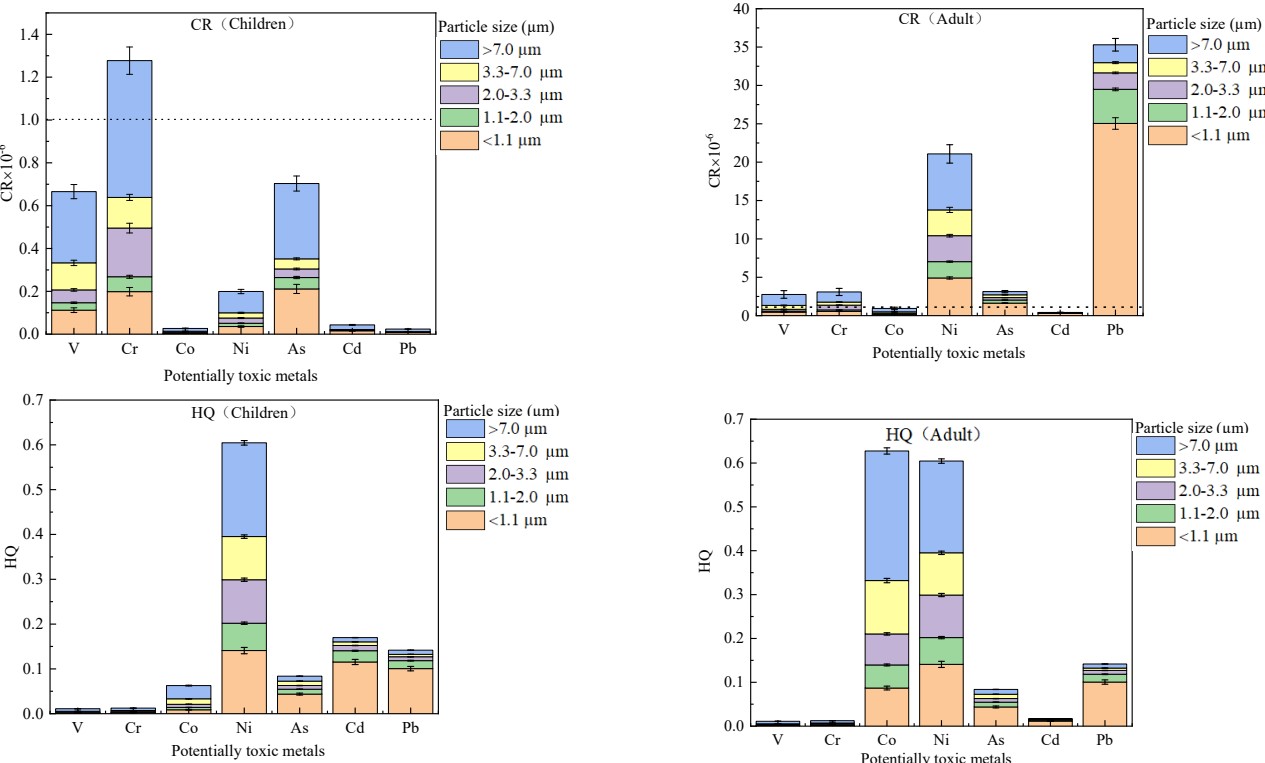

**Figure 9.** The carcinogenic and non- carcinogenic risks of toxic elements within $PM_{1.1}$, $PM_{1.1-2.0}$, $PM_{2.0-3.3}$, $PM_{3.3-7.0}$, $PM_{>7.0}$ in Beijing during the sampling period.

With respect to children and adults' non-carcinogenic risk, the corresponding contributions of elements to the HQ were ranked in the following order: Cr (VI) (46.06%) > V (28.42%) > As (16.71%) > Ni (6.28%) > Co (1.05%) > Cd (0.95%) > Pb (0.53%) and Pb (52.92%) > V (31.59%) > As (4.69%) > Cr (VI) (4.63%) > V (4.13%) > Co (1.41%) > Cd (0.63%). The HQ values for As, Cd, Co, Cr (VI), Ni, and V via inhalation exposure for both children and adults were all lower than the safe level (=1), indicating no non-carcinogenic risks from the inhalation exposure for each toxic elements [41].

## 4. Conclusions

In this study, nine water-soluble ionic species and 21 elemental species in PM which collected by Andersen high-flow five-stage sampler in the largest residential in Asia

in Beijing and the mass concentrations, size-distributions, and health risk assessment were characterized.

(1) The mass concentration of atmospheric particulate matter is positively proportional to ambient temperature and relative humidity, and negatively related to the wind direction. The sum of the mass concentrations of $PM_{1.1}$ and $PM_{1.1-2.0}$ was above the standard recommended by the WHO and NAAQS of $PM_{2.5}$.

(2) The mass concentration of $SO_4^{2-}$, $NO_3^-$, and $NH_4^+$ have demonstrated a unimodal distribution in fine particles, $PM_{1.1}$. The average mass ratios of $(NO_3^- + NO_2^-)/SO_4^{2-}$, $Cl^-/Na^+$, $Cl^-/K^+$, and $Cl^-/(NO_3^- + NO_2^-)$ were 1.68, 6.58, 6.18, and 0.57, respectively. Combined with higher CEFs, it showed that coal combustion and vehicle emissions were the main anthropogenic sources of PM in Beijing in winter.

(3) $PM_{1.1}$ was the major contributor of Pb, Cd, and As for CR and HQ. The potentially toxic metals of Cr (VI), V and As caused higher CR for children than Ni, Cd, Co, and Pb. Meanwhile, Pb and Ni were the cause of higher CR for adults than As, Cr (VI), V, Co, and Cd, especially in $PM_{1.1}$.

Our results can help stakeholders and policy makers to recognize the characteristics of anthropogenic particles and their impact on air quality in the region, and initiate strategies to further control emissions to improve public health. We recommend continuing efforts in controlling coal burning throughout the year and also to include the surrounding areas.

**Supplementary Materials:** The following are available online at https://www.mdpi.com/2227-9717/9/3/552/s1, Fig S1: The relative portions of carcinogenic and non- carcinogenic risks of toxic elements within $PM_{1.1}$, $PM_{1.1-2.0}$, $PM_{2.0-3.3}$, $PM_{3.3-7.0}$ and $PM_{>7.0}$ in Beijing during the sampling period, Table S1: Each group represents the time series of average ambient temperature (AT; °C), relative wind speed (WS; km/h) and wind direction(WD) in Beijing from 2018 December 26 to 2019 January 11, Table S2: Detailed information concerning the meteorological parameters during the sampling period is presented, Table S3: The P value of Pearson correlation coefficients between mass concentration and meteorological parameters and atmospheric pollutants, Table S4: Classification of Pearson correlation coefficients, Table S5: Detailed information about the concentration of different size-resolved PM ratios, Table S6: The size distribution of elements for Beijing samples during the winter, Table S7: The non- carcinogenic risks of toxic elements for children and adult by inhalation route, Table S8: The carcinogenic risks of toxic elements for children and adult by inhalation route.

**Author Contributions:** Conceptualization, K.X., A.Q., and Q.W.; methodology, S.L., W.W., and A.Q.; software, K.X. and A.Q.; validation, W.W. and S.L.; formal analysis, K.X. and Q.W.; investigation, A.Q.; resources contribution, Q.W.; data curation, K.X. and Q.W.; writing—original draft preparation, K.X.; writing—review and editing, K.X., S.L., and Q.W.; visualization, A.Q.; supervision, K.X. and Q.W.; project administration, Q.W.; funding acquisition, Q.W. All authors have read and agreed to the published version of the manuscript.

**Funding:** This study was partially supported by the Special Funds for Innovative Area Research (No. 20120015, FY 2008-FY2012) and Basic Research (B) (No. 24310005, FY2012-FY2014; No.18H03384, FY2017~FY2020) of Grant-in-Aid for Scientific Research of Japanese Ministry of Education, Culture, Sports, Science and Technology (MEXT) and the Steel Foundation for Environmental Protection Technology of Japan (No. C-33, FY 2015-FY 2017).

**Acknowledgments:** Some works of this study were supported. I would like to express my gratitude to Shinichi Yonemachi for helping us test potentially toxic metals via ICP-MS.

**Conflicts of Interest:** The authors declare that they have no conflict of interest.

## Abbreviations

| | |
|---|---|
| Airborne particulate matter | PM |
| Average lifetime | ATn |
| Beijing-Tianjin-Hebei | BTH |
| California Environmental Protection Agency | CALEPA |
| Carcinogenic risks | CR |
| Center for Environmental Science in Saitama | CESS |
| Crustal enrichment factors | CEFs |
| Exposure concentration | EC |
| Exposure time | ET |
| Exposure frequency | EF |
| Exposure duration | ED |
| Hazard quotient | HQ |
| High-volume air sampler | HV-RW |
| Inductively coupled plasma mass spectrometry | ICP-MS |
| International Agency for Research on Cancer | IARC |
| Integrated Risk Information System | IRIS |
| Ion Chromatography | IC |
| National Bureau of Statistics | NBS |
| Polytetrafluoroethylene | PTFE |
| Provisional Peer-Reviewed Toxicity Values | PPRTVs |
| Risk Assessment Information System | RAIS |
| Relative humidity | RH |
| Reference concentration | RfC |
| Temperature | T |
| Total suspended particles | TSP |
| Unit risk | UR |
| Ultrafine particles | UFPs |
| United States Environmental Protection Agency | USEPA |
| Wind speed | WS |
| Wind direction | WD |
| Water-soluble inorganic ionic species | WSIIs |
| World Health Organization | WHO |

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
