# Peer review of "Study on the Characteristics of Size-Segregated Particulate Water-Soluble Inorganic Ions and Potentially Toxic Metals during Wintertime in a High Population Residential Area in Beijing, China"

_processes, doi:10.3390/pr9030552_

Round 1

Reviewer 1 Report

The manuscript “Study on the characteristics of size-segregated particulate water-soluble inorganic ions and metal elements during wintertime in a high population residential area in Beijing, China” provides a number of information that may be interesting for the scientific community. Information processing is done well. The manuscript should undergo moderate English editing, please address this during revision. Therefore, without this clarification, it is difficult for me to recommend the manuscript for publication in its present form in processes.

please for the references respect the journal guidelines

Heavy metals: it is best to use potential toxic metals (i.e. 10.1016/j.ecoenv.2017.11.041 ;  10.1007/s00128-019-02605-1 ; 10.1016/j.jenvman.2017.11.080)

introduction

PM10: 10 is subscript

 highlight the novelty of the work

  1. Materials and Methods: the section number is 2

Figures 1 and 2: are cut out

"The aerosol sampling was conducted by a high-volume air sampler at a flow rate of
566 L/min in Beijing from 26 December 2018 to 11 January 2019. The sampler (Anderson
Sampler, equipped with 5 cut size :1.1, 1.1- 2.0, 2.0-3.3, 3.3-7.0, 7.0 μm) can collect the particulate matter in the flue gas on five quartz filter membranes according to the aerodynamic diameter. The quartz filter used for collecting PM was baked in a muffle furnace
(450 ℃) for 6 hours before sampling and placed in a constant temperature and humidity
chamber at 25 ℃ and 45 % humidity for 24 hours and then wrapped in clean aluminum
foil paper and placed in a refrigerator at −45 ° C until use. After sampling, the membranes
were equilibrated and weighed again using the same procedure. To ensure accuracy, each
filter was weighed at least three times before and after sampling, and the results were
averaged. After weighing, store the filter at −20 ° C until analysis. During the sampling
period, the temperature (T; ℃), relative humidity (RH; %), wind speed (WS; km/h) and" please insert the standard methods and procedures

"1.2. Elemental characterization via ICP-MS
1.2. Water-soluble inorganic ions"  : please number paragraphs correctly and insert the standard methods and procedures

"21 elemental
species (Na, Mg, Al, K, Ca, Ti, V, Cr, Mn, Fe, Co, Ni, Cu, Zn, As, Se, Sr, Cd, Sb, Ba, Pb) "  why just these elements?

2.6:   write the analyses  carried out

"2. Results and Discussion " section number 3

3.1: The results are well represented, however there is no discussion. For example, how meteorological factors influence the diffusion of particles. There don't seem to be any trends.

Figs 3 and 5: there are no standard deviations?

Reviewer 2 Report

This article deals with the current and acute topic of air pollution of the very large city of Beijing by particulate matter (PM) extended by the presence of bound heavy metals and other substances and the evaluation of their impact on human health. The amount of PM is described by the weights of the PM fractions, which is a somewhat outdated approach, because particles numbers rather than weights is currently more accurate description, but in this case, it does not matter so much, because the benefit of the article is mainly to directly identify, quantify and determine the sources of pollution of PM and bound substances and their impact on human health. Also, different calculations are made to account for children and adults. The Conclusions are well arranged and give a clear direction for priorities to protect the air quality in the monitored area.

The main reservation for this article is that the results obtained are unfortunately severely limited by the short observation period of 16 days in the winter season. The explanation of why such a short period was chosen and how this may affect the overall picture of PM air pollution in the monitored area should be given in more detail in the text.

There are other reservations about the article, mainly regarding the legibility of the text, the English language, and others, which are summarized below, but if these are corrected, this article should be suitable for publication.

Comments and suggestions:

  1. A more detailed explanation in the context of the researched area, why the measurement was performed only in such a limited period of time.
  2. The English language needs to be improved.
  3. Many abbreviations are used. A nomenclature is necessary for clarity.
  4. Section 2.4 below Table 1 is vaguely described. There is talk of HQ and CR and their influence, but these values ​​are not in Table 2. The results are then in section 3.4.3. Would it be possible to consolidate these two sections into one for clarity?
  5. Section 2. Results and Discussion should be section 3.
  6. Section 3.1. In addition to referring to Table 1 instead of Table 3 and the values ​​of 2.18 + - 2.55 °C are the same as those in section 2.1 (which is correct?), there is no connection to the rest of the text. An explanation should be added that will create such connection.
  7. Section 3.2. The value at which fine particles peak is missing.
  8. The text under Fig. 5c. is difficult to read due to many numbers. It would be better to write it in a table.
  9. Section 3.4.3. Same as point 8.
  10. Figure 7. Add a limit line for CR at 1x10-6. For HQ, extend the scale to 1, which is the limit value.

Reviewer 3 Report

In the  manuscript I noticed minor remarks that should be taken into account before possible publication

Comments

  1. The file should have the line number so the reviewer can mention the line number for each point.
  2. All citations in the text should be reformat and follow the MDPI journal's instruction for citations
  3. No reference in the text to Figure 1 and 6
  4. Page 3. 1. Sites and sampling - should be 1.1
  5. Page 4. 2.2. After weighting, A certain…., should be After weighting, a certain
  6. Page 4 2. Water-soluble inorganic ions – sholuld be 1.3

Please correct the numbering of subsections

  1. Page 5. 4., 4th row: Health risk assessment.

Consistently, as for Group 1 and 2 B, and for Group 2A, name them what they are.

Group 2A (probably carcinogenic to humans):…

  1. Unify the same format, e.g.. mg/m3 or mg×m-3; description of figures - full name or abbreviation, Figure 2.  or    3.
  2. Page 7: Statistics – please describe which methods and significance levels were used
  3. The references should be reformat and follow the MDPI journal's instruction for references preparation. 
  4. Page 2: Hu, Xin, et al., 2012; page 12: Jin Zhang et al., 2017; page 111: Al-Humour et al., 2019 - not in References

Please check and complete

Round 2

Reviewer 1 Report

The authors responded adequately to all comments. It seems strange to me not to see standadrd errors in the graphs.

Reviewer 2 Report

The authors reviewed the text accordingly, but some objections persist.

Major:

  1. The text is still missing an explanation as how such a short sampling period of 17 days can be representative enough to make general conclusions presented by the paper.

Minor:

  • Line 104 a Line 232. It is a coincidence that the mean temperature over the entire year is the exact same as the mean temperature of the sampling period?

Also, ± -value is not a valid notation.

  • Line 258, Line 270, Line 285 and Line 350. Consider using tables for better readability.
  • Line 325. Use same text formatting for entire article.
  • Line 199. “An HQ value higher than one implies…“ Instead of “one“ use the same numbering format as in the rest of the paragraph: “1“ or “1x10^0“
